# Impact of Self-Efficacy and Perfectionism on Academic Procrastination among University Students in Pakistan

**DOI:** 10.3390/bs13070537

**Published:** 2023-06-28

**Authors:** Muhammad Azeem Ashraf, Namood-e Sahar, Muhammad Kamran, Jan Alam

**Affiliations:** 1Research Institute of Education Sciences, Hunan University, Changsha 410082, China; azeem@hnu.edu.cn; 2National Institute of Psychology, Quaid e Azam University, Islamabad 04403, Pakistan; namood.sahar@nip.edu.pk; 3Department of Education, University of Loralai, Balochistan 85200, Pakistan; muhammad.kamran@uoli.edu.pk; 4Department of Education, University of Wah, Rawalpindi 47010, Pakistan

**Keywords:** self-efficacy, perfectionism, academic procrastination, higher education, Pakistan

## Abstract

This study aimed to evaluate the impact of self-efficacy and perfectionism on academic procrastination among university students and its differences among genders in Pakistan. It was hypothesized that self-efficacy and perfectionism would significantly impact academic procrastination and that there is a significant difference in students’ views concerning their gender. The sample comprised 405 university students, 104 male and 301 female. The study used the general self-efficacy scale, the multidimensional perfectionism scale, and the academic procrastination scale to measure the constructs. SmartPLS 4 was applied for the analysis of the data. The results indicated that all three variables—self-efficacy, perfectionism, and academic procrastination—were present among university students. Perfectionism showed a significant effect on academic procrastination. However, self-efficacy showed no significant effect on academic procrastination. Further, no significant difference was found in students’ views concerning their gender. The findings provide significant evidence for stakeholders to improve academic procrastination among university students.

## 1. Introduction

The term procrastination emerged from the Latin words pro, which means “forward or onward, presumptuous, or in favor of,” and crastinus, suggesting “of tomorrow” [1]. Procrastination unnecessarily puts off everyday jobs to the point that one starts to feel uneasy [2]. Procrastination can be permanent or temporary and can be defined as a rationale for delaying behavioral or cognitive output, making a decision, or taking an action [3]. Academic procrastination involves failing to carry out an activity within the needed duration or delaying the task until the last-minute performance one eventually intends to accomplish [4]. Academic procrastination can lead to failure to achieve academic goals on time, resulting in the progression of emotional distress in individuals [5,6]. It also leads to incompetent behavioral consequences, and a person may have problems dealing with their surroundings effectively [7].

University students that procrastinate seem to delay and postpone their academic work, becoming self-explanatory and disregarding their academic obligations during their whole study period [8]. It appears to be a widespread tendency for university students to put off their academic work, which causes delays in finishing projects, preparing for exams, and submitting assignments and presentations [9]. Academic procrastination denotes a delay in academic activity in education and training [10]. It may be deliberate, unintentional, or habitual, but it substantially impacts university students’ learning and success [11].

Cognitive and behavioral factors that favor or hinder student performance in their academic activity and how they relate to academic success have stirred interest in educational psychology [12]. Of these factors, self-efficacy, defined by Bandura [13] as the personal capacity to cope with specific situations, is a determining psychological variable strongly predictive of academic achievement [14,15]. This is because self-efficacy involves beliefs about one’s capacity to organize and execute actions to achieve specific results [12].

Perfectionism is setting up higher values or standards by an individual for doing a task that is critically evaluated by the individual [16,17]. Adler was the first to describe the theory of perfectionism. He argued that the struggle for perfectionism is an inborn ability and is considered normal because of the propensity of human beings. However, he argues the difference between healthy perfectionism, which involves goals that are obtainable, and maladaptive perfectionism, which leads to obsessive order and fear of critique. High personal standards are not always problematic [18]. According to the Canadian Psychological Association, perfectionism is a personality trait linked with more interpersonal, emotional, and success-related problems. It is not considered a disease, but it is a susceptibility factor that can create difficulties in the lives of adults [19]. Many researchers considered perfectionism multidimensional rather than a unidimensional construct [16]. Perfectionism is a multidimensional construct with interpersonal and personal aspects [20]. The three main dimensions of perfectionism are other-oriented perfectionism (OOP), self-oriented perfectionism (SOP), and socially prescribed perfectionism (SPP) [21].

Self-oriented perfectionism (SOP) is the motivational force for setting high standards for oneself, and the assessment of one’s behavior is based on these high standards [22]. Perfectionism directed at and from oneself is referred to as self-oriented perfectionism [23]. It involves establishing high criteria associated with productivity and success in a career [24]. This element of perfectionism entails a critical assessment of oneself, a focus on flaws and errors, and a greater drive to achieve perfection [21]. Other-oriented perfectionism (OOP) is setting high, idealistic standards for others and then harshly evaluating them in light of these standards [25]. Lack of trust, blame that is other-focused, and feelings of hostility against other people might come from others not living up to these expectations [26]. It involves having high and unrealistic expectations of other people [21]. Achieving the expectations and goals established for them by other people is known as socially prescribed perfectionism (SPP) [27,28]. It is the idea that one should fulfill other people’s expectations. It includes being conscious of the expectations others have of you because you are a perfectionist [20].

The present study aimed to check the impact of self-efficacy and perfectionism on academic procrastination among university students. This study investigated to what extent self-efficacy and perfectionism cause students to show academic procrastination. Research has been undertaken on these variables separately or with other variables. The present study was conducted to probe and scrutinize the relationship between these three variables, i.e., self-efficacy, perfectionism, and academic procrastination. Each of these three variables is assumed to have a great impact on young adults. This is because academic procrastination poses a hindrance to success for students. So, the present study highlights some factors related to procrastination that will help students identify some causes. The main objectives of the study include:To investigate self-efficacy, perfectionism, and academic procrastination among university students;To find the effects of self-efficacy and perfectionism on the academic procrastination of university students;To determine differences in the respondents’ views concerning their gender about the effects of self-efficacy and perfectionism on academic procrastination.

### 1.1. Students’ Self-Efficacy towards Academic Perfectionism

In recent years, a growing emphasis has been placed on students’ ability to control their academic learning and performance [29]. Academically, students’ self-regulation is stimulated by metacognition, behavior, and motivation in their learning process [30]. Social cognitive theorists believe self-regulation comes after setting goals [31], which reveals a strong sense of efficacy in skills [32]. Setting up proximal goals enhances self-efficacy for academic achievement and arouses internal interest [33] because self-regulated learners use strategies [34]. Student self-monitoring and performance evaluation are prompted by setting goals [35]. Social cognitive theorists claim self-judgment and self-monitoring are outcomes of setting goals [36]. At the same time, motivation and self-regulation get support from setting self-efficacy and personal goals [37]. Self-efficacy affects the setting of goals, efforts, and persistence of a person [38], which directly or indirectly affects performance accomplishment [39]. Organizational goals involve personal goals, indicating parental expectations’ effects on children’s achievements [40]. In the light of social cognitive theory, students’ self-efficacy and parental expectations affect their goals [41]. Students’ class work, conducive environment development, and class participation are all outputs of self-efficacy [42], which help in academic achievement. Self-regulation leads to self-efficacy, which develops personal goals and academic/grade achievements that are components of perfectionism.

### 1.2. Self-Efficacy Beliefs and Students’ Engagement in Learning

Students’ motivation and learning are of the highest concern at all levels of education and are a chronic problem for all educationists [43]. Students are motivated when they know a topic or course is interesting, exciting, and important [44], leading to student engagement in learning [45]. Students’ motivations are directly concerned with their beliefs and capabilities in studies, which are the output of their self-efficacy [46], which can be perceived as self-concept and competence [47]. From self-efficacy to learning and achievement, it requires three basic mediators [48]. The first mediator is behavioral engagement, which requires students’ efforts, persistence, and instrumental support, from self-efficacy to learning and achievement [42]. The second mediator is cognitive engagement, which is supported by students’ learning strategies and metacognition [49]. The third mediator is motivational engagement, supported by interest in learning, valuing the topic or course, and its effects [50]. Strong self-efficacy beliefs help with persistence while completing a task or learning [51]. Self-doubt does not permit a student to complete their task, even if it is easy to perform [52]. Students with low academic self-efficacy rarely ask for help as they are not exposing their weaknesses to others, which is quite the opposite of high self-efficacy holders [53]. Students’ self-efficacy is positively related to their behavioral engagement [54]. While cognitive engagement allows students to continue cognitively in their learning process [55]. Because there may be occasions where the students are quite confident with their previous knowledge, which stimulates learning engagement [56], self-efficacy is based on a motivational construct that requires personal interest and the belief that the task or learning is important for the learner [57]. Engaging students in an activity improves their skills, knowledge, and experience to develop beliefs [58].

### 1.3. Procrastination among University Students

Students’ productivity and work accomplishment are important in almost all societies [59]. However, procrastination directly disrupts these norms, resulting in delays in work accomplishment and decision-making that lead to anxiety among students [60]. Procrastination has internal and external consequences. Internal consequences consist of self-blame, gloom, repentance, and frustration [61]. External consequences are costly and consist of weak academic and work progress, lost opportunities, and stressed relationships [61]. Research has proven that more than 50% of university students face frustration and anxiety due to procrastination in their assignment writing [62], while doctoral students discontinue their research and dissertation as a result of their procrastination [63]. Behaviorists connect procrastination to human preferences [64]. At the same time, psychodynamic theorists consider procrastination a rebellion against many demands [65]. Researchers consider unreasonable beliefs, ascription style, time importance, and self-esteem as major predictors of procrastination [66]. It is a strategy for protecting self-esteem that allows for actual performance under personal circumstances [67]. Bandura explained self-efficacy as an individual’s judgment while performing in certain situations [68]. Weak self-efficacy leads to procrastination [69].

### 1.4. Theoretical Framework and Hypotheses

The relationship between self-efficacy, perfectionism, and academic procrastination could also be elaborated through theoretical frameworks. Lazarus and Folkman’s appraisal-anxiety theory proposes that the cognitive appraisal attached by an individual to task accomplishment determines the development of procrastination. If the task is appraised as beyond an individual’s capacity to accomplish (i.e., low self-efficacy), it will result in procrastination. It was also suggested by the theory that anxiety also plays an important contributory role in decreasing an individual’s self-efficacy, which might be because of a high ratio of past failures [70]. Likewise, when a student experiences repeated failures or fears, it might decrease their self-efficacy, increasing academic procrastination. Thus, the student would keep on leaving the academic assignments and tasks until the eleventh hour. On the contrary, students with a high sense of confidence in their abilities and capabilities to effectively accomplish the tasks would procrastinate less.

The control theory of self-regulation by Sirois et al. also attempts to explain the relationship between our study variables. According to the theory, perfectionism is signified by high standards for achieving any goal and high doubts about oneself for accomplishment, i.e., low self-efficacy. The perfectionist thus perceived their current state and the desired one as having many discrepancies they could not address, which eventually resulted in high procrastination [71]. In the present study, students with low self-efficacy regarding their academic tasks are unable to accomplish them. Also, high perfectionism is associated with low self-efficacy. Thus, students with high standards of perfectionism and low self-efficacy might have a high level of procrastination. Thus, the study aims to examine the following hypotheses:There is a significant effect of self-efficacy and perfectionism on the academic procrastination of university students.There is a significant difference in university students’ views concerning their gender about the effects of self-efficacy and perfectionism on academic procrastination.

## 2. Research Design

In the present study, a correlational survey research design was used as the study’s primary aim was to assess the relationship between self-efficacy, perfectionism, and academic procrastination.

### 2.1. Sample

The present study sample consisted of 405 students enrolled in colleges and universities located in four provinces (Punjab, Sindh, Khyber Pakhtunkhwa, and Balochistan) and one semi-province (Gilgit-Baltistan) of Pakistan. Among all participants, 35 enrolled in Intermediate, 77 enrolled in Bachelors, 162 enrolled in a Master’s, 102 enrolled in an MPhil, and 29 enrolled in a Ph.D. Regarding gender, 104 were male, while 301 were female. Most students (67%) were between the ages of 20 and 25. Table 1 provides the participants’ information.

### 2.2. Instruments

#### 2.2.1. General Self-Efficacy Scale

This study used the general self-efficacy scale developed by Schwarzer and Jerusalem [72]. The scale consisted of 10 items that measure self-efficacy on 4-point Likert-type response patterns ranging from 1 (not at all true) to 4 (exactly true). Low scores indicate low levels, and high scores indicate high levels of self-efficacy. Scores can range from 10 to 40. The study has reported the alpha reliability of academic procrastination as (alpha = 0.9.), perfectionism as (alpha = 0.823), and self-efficacy as (alpha = 0.632) in Table 2.

#### 2.2.2. Multidimensional Perfectionism Scale

In the present study, the multidimensional perfectionism scale developed by Hewitt et al. was used to measure perfectionism [73]. It has a total of 45 items divided into three sub-scales: self-oriented (SO), other-oriented (OO), and socially prescribed (SP). Every sub-scale consists of 15 items. A high score on any subscale represents a propensity to be a perfectionist on that measured aspect. The responses are given on a 7-point Likert-type scale ranging from 1 (strongly disagree) to 7 (strongly agree). The Items 2, 3, 4, 8, 9, 10, 12, 19, 21, 24, 30, 34, 36, 37, 38, 43, 44, and 45 of the scale were scored reversely. Hewitt and Flett reported a coefficient alpha of 0.86 for self-oriented perfectionism, 0.82 for other-oriented perfectionism, and 0.87 for socially prescribed perfectionism [74].

#### 2.2.3. Academic Procrastination Scale

In the present study, the academic procrastination scale developed by McCloskey and Scielzo was used to measure academic procrastination [75]. The scale consisted of 25 items on a 5-point Likert-type response pattern ranging from 1 (disagree) to 5 (agree). Low scores indicate a low level, and high scores indicate a high level of academic procrastination. The scores on the scale range from 25 to 125. Item numbers 1, 8, 12, 14, and 25 on the scale were reversed score items on the scale.

### 2.3. Ethical Consideration

In regard to research ethics and maintaining the study’s quality, ethical considerations were followed with the greatest attention. The Ethics Review Committee of the University of Loralai approved the study. Before participation in the study, participants’ consent was obtained. Moreover, participants were given the choice to withdraw at any time. All participants were briefed before participating in the questionnaire. The confidentiality and privacy of the participants’ information were maintained. No harm was caused to any animal or human throughout the study.

### 2.4. Procedure

A questionnaire was developed online, containing background information about the participants and the main instruments. After approval from relevant departments, the questionnaire was sent to student management offices in different universities, and these offices forwarded the questionnaire to the students. Before participation in the questionnaire, consent was obtained from all participants. Detailed instructions were given to fill out each questionnaire. All the participants were thanked for their precious time and full cooperation.

## 3. Results

The partial least squares structural equation model (PLS-SEM) via SmartPLS 4.0 was applied to analyze the data. The analysis used a measurement model for internal consistency, reliability, discriminant validity, and convergent validity. In the structural model, R square and path coefficient were performed along with multigroup analysis.

Convergent validity and internal consistency reliability are summarized in Table 2. All the constructs and indicators met reflective measurement criteria. As outer loadings (λ), the indicator reliability is achieved as (factor loading > 0.511) at the minimum level. The values for average (AVE) are (AVE > 0.193) at the minimum level, which shows the convergent validity as achieved. Composite reliability (CR) is (CR > 0.688) at the minimum level, which is the required value for internal consistency. The reliability indices for each construct are represented by rho-A (rho-A > 0.579). To highlight the collinearity of the items, the variance inflation factor (VIF) is indicated as (VIF > 1.118) at the minimum level. The descriptive statistics of all the items are indicated via mean and SD in Table 2. The mean score of almost all the items is (mean > 2.29, SD < 1). The mean score highlights the exitance of self-efficacy, perfectionism, and academic procrastination among university students in Pakistan. All the details in the table indicate that measurement criteria have been met.

A construct is made unique compared to its counterparts by its discriminant validity, which determines whatever the author proposes. The discriminant validity must be greater than all others in columns and rows in its line [76]. As highlighted in Table 3, the diagonal values are greater than all their counterparts, indicating discriminant validity.

The structural model in Table 4 highlights R square, the sample mean, t statistics, and *p* values. The collinearity of the model was tested before the structural model highlighted in Table 2 via the variance inflation factor (VIF). A significant positive effect of perfectionism on academic procrastination is highlighted as (β = 0.452, T value = 11.319, *p* = 0.000, *p* < 0.001). According to this study, self-efficacy shows no significant effect on academic procrastination since (β = −0.030, T values = 0.362, *p* = 0.717, *p* > 0.05), as shown in Table 4.

Details of the path coefficient for the effects of perfectionism and self-efficacy are highlighted in the histograms in Figure 1 and Figure 2, respectively. All values are shown to be highly significant and correlated, which ensures the existence of self-efficacy and perfectionism among university students in Pakistan. The bell-shaped histograms indicate the normality of the research data.

The data was passed through multiple group comparisons to find differences in students’ views concerning their gender. The sample size of male and female students was not equal, so the Welch test was applied to find differences. The detail in Table 5 for the effects of perfectionism on academic procrastination is as follows: (difference = 0.147, t value = 1.845, *p* value = 0.067, *p* > 0.05), which indicates no significant difference in students’ views concerning their gender as male and female. The detail of self-efficacy is (difference = 0.139, t value = 0.979, *p* value = 0.329, *p* > 0.05), indicating no significant difference.

## 4. Discussion

The study extends the present literature by examining the effects of self-efficacy and perfectionism on academic procrastination among university students in Pakistan. Further, it also investigated this impact regarding gender differences. The general self-efficacy scale, multidimensional perfectionism scale, and academic procrastination scale were applied to measure self-efficacy, perfectionism, and academic procrastination.

This study’s first objective was to investigate self-efficacy, perfectionism, and academic procrastination among university students. The findings of the descriptive analysis in Table 2 confirm that all these variables (self-efficacy, perfectionism, and academic procrastination) were present. These findings are consistent with the findings of Kurtovic et al. about the existence of the three factors in educational institutions [77].

The second objective of this study is to find the effects of self-efficacy and perfectionism on the academic procrastination of university students. The findings indicate a highly significant impact of perfectionism on the academic procrastination of university students in Pakistan. Several studies [20,21,25] in the past provide similar evidence that perfectionism significantly impacts academic procrastination. Thus, the findings of this study are consistent with the previous literature.

Self-efficacy was found insignificant in its impact on academic procrastination in this study. These findings are consistent with the results of Honmore and Jhadav [78] on the effects of self-efficacy on academic procrastination among students. One possible reason is that family background and environment may influence students’ self-efficacy and academic procrastination [79]. The findings of this study are also in line with Ede, Sullivan, and Feltz’s [52] finding that self-doubt and incompetency do not allow students to complete a task, leading to academic procrastination.

The study’s third objective was to find a gender difference among the impacts of perfectionism and self-efficacy on academic procrastination. The findings show no significant differences among students’ genders in the effects of self-efficacy and perfectionism on academic procrastination. These results are similar to Ghosh and Roy’s findings that there is no significant difference in students’ opinions concerning their gender about self-efficacy, perfectionism, and academic procrastination [80]. A similar study in the Pakistani context on the analysis of procrastination based on workload among university students in Pakistan indicated no gender difference among students [11]. Bandura explained that self-efficacy is a personal capacity that depends on context and situation [13]. The position and context for both male and female students are the same in relation to their academic procrastination as caused by self-efficacy and perfectionism.

## 5. Implications and Limitations

The results of this study offer an excellent understanding for different stakeholders (i.e., psychologists, teachers, policymakers, and students) in educational institutions about students’ academic procrastination. Perfectionism and self-efficacy must be considered when assessing students’ academic procrastination. The results help understand students’ perceptions of their self-beliefs and how these beliefs can affect their whole personalities and influence them to achieve their goals. It is helpful for students’ awareness to control external or internal happenings and can also interpret their perfectionism. Moreover, this study provides authentic ways for stakeholders to provide fruitful information and understand particular problems related to students’ beliefs about their capacities and potential to perform well. Furthermore, as self-efficacy decreases procrastination, a focus should be placed on supporting students in improving their self-efficacy. One way of doing this is by helping students set goals to achieve desired outcomes [81]. Setting goals for students helps them realize their main goals in the learning process and thus improve their self-efficacy.

However, this study’s findings also have some limitations that must be reported. First, this study used a self-reported questionnaire to collect the data, which may have an element of social acceptability among participants while providing their answers. Future studies may consider a mixed-methods approach to minimize this limitation. The second limitation is the generalizability of the data, as the data for this study were collected from university students only. Future studies should collect data from different populations, such as schools or colleges, to increase the study’s generalizability. The third limitation is the selection of instruments. This study used the general self-efficacy scale, the multidimensional perfectionism scale, and the academic procrastination scale to measure self-efficacy, perfectionism, and academic procrastination. Future studies may consider using different scales to measure these variables.

## Figures and Tables

**Figure 1 behavsci-13-00537-f001:**
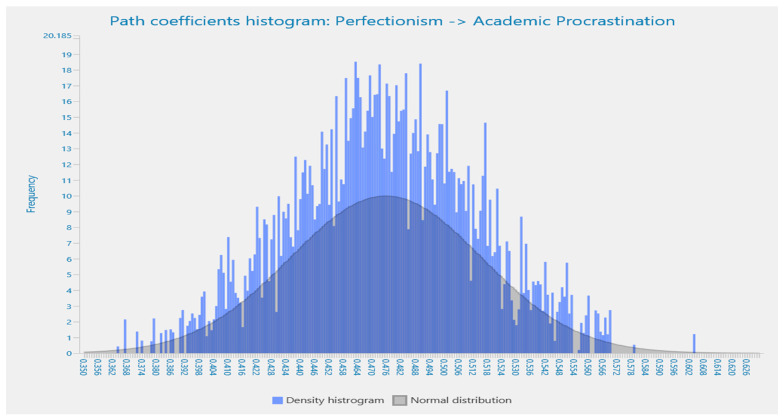
Path coefficient histogram for the effects of perfectionism on academic procrastination.

**Figure 2 behavsci-13-00537-f002:**
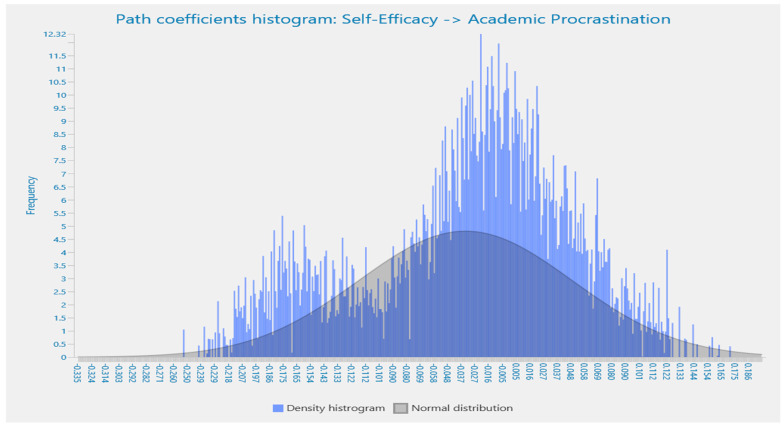
Path coefficient histogram for the effects of self-efficacy on academic procrastination.

**Table 1 behavsci-13-00537-t001:** Demographic information of participants (*N* = 405).

Variables		Frequency	Percentage	Missing
Gender	Male	104	26%	
	Female	301	74%	
Age (in years)	20–25	270	67%	24
	25–30	44	11%	
	30–35	32	8%	
	Above 35	35	9%	
Province	Balochistan	44	11%	7
	Punjab	279	69%	
	Sindh	25	6%	
	Khyber Pakhtunkhwa	44	11%	
	Gilgit Baltistan	2	1%	
Education	12 years (Intermediate)	35	9%	
	14 years (BA/BSc)	77	19%	
	16 years (MA/MSc/BS)	162	40%	
	18 years (MS/MPhil)	102	25%	
	Above 18 years (PhD)	29	7%	
Study Discipline	Business	18	4%	6
	Humanities	71	18%	
	Natural Sciences	97	24%	
	Social Sciences	213	53%	
Area of Residence	Rural	130	32%	11
	Urban	264	65%	

**Table 2 behavsci-13-00537-t002:** Measurement model.

Items	Factor Loading	Cronbach’s Alpha	Rho A	CR	AVE	Mean	SD	VIF
Academic Procrastination		0.9	0.901	0.913	0.346			
AP10	0.586					2.506	0.048	1.653
AP11	0.639					2.395	0.041	1.834
AP13	0.569					2.291	0.042	1.507
AP15	0.682					2.336	0.032	2.041
AP16	0.511					2.501	0.046	1.451
AP17	0.674					2.230	0.036	1.766
AP18	0.493					2.457	0.052	1.526
AP19	0.643					2.442	0.041	1.935
AP2	0.538					2.731	0.041	1.562
AP20	0.641					2.565	0.041	1.793
AP21	0.604					2.659	0.043	1.656
AP22	0.632					2.556	0.041	1.762
AP23	0.622					2.514	0.038	1.607
AP24	0.556					2.496	0.047	1.555
AP3	0.531					2.768	0.045	1.516
AP4	0.552					2.546	0.049	1.457
AP5	0.502					2.746	0.046	1.531
AP6	0.534					2.674	0.048	1.725
AP7	0.617					2.701	0.038	1.881
AP9	0.591					2.610	0.041	1.638
Perfectionism		0.823	0.818	0.849	0.193			
MP10	0.436					2.642	0.061	1.343
MP12	0.395					3.235	0.064	1.261
MP13	0.472					2.963	0.056	1.277
MP18	0.376					2.575	0.065	1.346
MP19	0.472					2.923	0.051	1.351
MP20	0.461					2.933	0.061	1.477
MP22	0.544					2.691	0.049	1.391
MP23	0.368					2.842	0.067	1.319
MP24	0.374					2.963	0.061	1.176
MP25	0.529					2.642	0.045	1.358
MP26	0.471					3.000	0.062	1.503
MP27	0.385					2.830	0.067	1.365
MP31	0.477					2.514	0.051	1.329
MP33	0.468					2.795	0.058	1.394
MP34	0.432					2.580	0.057	1.491
MP35	0.386					2.746	0.062	1.429
MP37	0.452					2.909	0.057	1.221
MP39	0.518					2.768	0.056	1.488
MP41	0.384					2.993	0.067	1.269
MP45	0.405					2.728	0.059	1.224
MP5	0.501					2.630	0.042	1.235
MP6	0.276					3.067	0.076	1.469
MP7	0.407					2.719	0.067	1.349
MP8	0.459					2.731	0.062	1.484
Self-Efficacy		0.632	0.579	0.688	0.384			
SE10	0.578					3.126	0.32	1.309
SE2	0.897					2.852	0.465	1.118
SE7	0.558					2.852	0.321	1.255
SE8	0.294					2.998	0.322	1.281

**Table 3 behavsci-13-00537-t003:** Fornell–Larcker criterion.

	Academic Procrastination	Perfectionism	Self-Efficacy
Academic Procrastination	**0.588**		
Perfectionism	0.446	**0.439**	
Self-Efficacy	0.068	0.216	**0.621**

**Table 4 behavsci-13-00537-t004:** Structural model.

	Original Sample (O)	Sample Mean (M)	Standard Deviation (STDEV)	T Statistics (|O/STDEV|)	*p* Values
Perfectionism -> Academic Procrastination	0.452	0.476	0.040	11.319	0.000
Self-Efficacy -> Academic Procrastination	−0.030	−0.034	0.083	0.362	0.717

**Table 5 behavsci-13-00537-t005:** Gender Differences.

	Difference (Male–Female)	t Value (Male vs. Female)	*p* Value (Male vs. Female)
Perfectionism -> Academic Procrastination	0.147	1.845	0.067
Self-Efficacy -> Academic Procrastination	0.139	0.979	0.329

## Data Availability

The dataset for this study can be obtained from the corresponding author (janalam.jk@gmail.com) upon reasonable request.

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
