# Peer review of "Impact of Self-Efficacy and Perfectionism on Academic Procrastination among University Students in Pakistan"

_behavsci, 2023, doi:10.3390/bs13070537_

Round 1

Reviewer 1 Report

This was a very interesting study (N=405) and the statistical and factor analysis (incl t tests) were relevant, revelatory and well displayed.  There is a lot of data in the histograms presented, that could do with some more explanation for a lay reader.  For example, in simple terms, what were the average scores for each of the scales measured -ie. what did the sample get for self-efficacy score?  Procrastination score? etc.  I can see the variables and how they were correlated but would like to see some simple descriptive statistics too, for a more lay reader.

However, that said, a very interesting study.  Timely too, given the increase in anxiety being experienced by this college generation - and nice to know how this impacts on procrastination.

I would like to see the Discussion section added to - it repeats the data back to us and needs more nuance and explanation of where these findings sit in relation to CURRENT literature on these 3 topics.  It is currently only 3 small paragraphs and detracts from the writing, in its current form.

Author Response

Dear Reviewer,

We thoroughly read the manuscript and whatever changes you suggested we followed that for the final version. We are much thankful for your precious comments which helped us to learn more about our shortcomings. 

Best Regards!

Reviewer 2 Report

Thanks for this, I really enjoyed reading it.

Please see attached document. There are a series of minor corrections - should take under an hour to complete.

Well done.

Generally excellent throughout, with some minor lapses - please see attached document.

Author Response

(The authors gave the same response as above.)
